# Association of birth and childhood weight with risk of chronic diseases and multimorbidity in adulthood

Yue Zhang [1,2,7], Yaguan Zhou[1,2,7], Yangyang Cheng[1,2], Rodrigo M. Carrillo-Larco[3], Muhammad Fawad [1,2], Shu Chen[4,5] & Xiaolin Xu [1,2,6 ✉]

## Abstract

**Background** Little is known about the relationship between early life body size and occurrence of life-course multiple chronic diseases (multimorbidity). We aim to evaluate associations of birth weight, childhood body size, and their changes with the risks of chronic diseases and multimorbidity.

**Methods** This prospective cohort study included 246,495 UK Biobank participants (aged 40–69 years) who reported birth weight and childhood body size at 10 years old. Birth weight was categorized into low, normal, and high; childhood body size was reported as being thinner, average, or plumper. Multimorbidity was defined as having two or more of 38 chronic conditions retrieved from inpatient hospital data until 31 December, 2020. The Cox regression and quasi-Poisson mixed effects models were used to estimate the associations.

**Results** We show that 57,071 (23.2%) participants develop multimorbidity. Low birth weight (hazard ratio [HR] 1.29, 95% confidence interval [CI] 1.26–1.33), high birth weight (HR 1.02, 95% CI > 1.00–1.05), thinner (HR 1.21, 95% CI 1.18–1.23) and plumper body size (HR 1.06, 95% CI 1.04–1.09) are associated with higher risks of multimorbidity. A U-shaped relationship between birth weight and multimorbidity is observed. Changing to be thinner or plumper is associated with multimorbidity and many conditions, compared to changing to be average.

**Conclusions** Low birth weight, being thinner and changing to have a thinner body size in childhood are associated with higher risks of developing multimorbidity and many chronic conditions in adulthood. Early monitoring and maintaining a normal body size in childhood could have life-course benefits for preventing multimorbidity above and beyond individual conditions.

## Plain language summary

Little is known about the relationship between childhood body size and the risk of developing more than one chronic disease later in life. Using data from the UK, we found that low birth weight, high birth weight, and being thinner or plumper than average during childhood were all associated with higher risks of developing more than one chronic disease in adulthood. In addition, changing body shape during childhood to be either thinner or plumper, was associated with being more likely to develop more than one chronic disease later in life. Our results highlight the importance of early monitoring and maintenance of average body size in childhood, as this might prevent the occurrence of chronic diseases later in life.

[1] School of Public Health and The Second Affiliated Hospital, Zhejiang University School of Medicine, Hangzhou, Zhejiang, China. [2] Key Laboratory of Intelligent Preventive Medicine of Zhejiang Province, Hangzhou, Zhejiang, China. [3] Hubert Department of Global Health, Rollins School of Public Health, Emory University, Atlanta, GA, USA. [4] Australian Research Council Centre of Excellence in Population Ageing Research (CEPAR), University of New South Wales, Sydney, Australia. [5] School of Risk & Actuarial Studies, University of New South Wales, Sydney, Australia. [6] School of Public Health, Faculty of Medicine, The University of Queensland, Brisbane, Australia. [7] These authors contributed equally: Yue Zhang, Yaguan Zhou. ✉email: xiaolin.xu@zju.edu.cn

The World Health Organization has endorsed a global target of a 30% reduction in low birth weight and no increase in the proportion of overweight children by 2025[1]. Birth weight is an indicator of infant intrauterine development and health[2], and current studies have also associated it with incidences of chronic diseases in adulthood or later life. A cohort study using data from 256,787 participants of UK Biobank observed that participants with LBW had higher risks of coronary heart disease, stroke, and heart failure in their later life[3]. Another study of 15,792 individuals aged 45–64 years in the Atherosclerosis Risk in Communities Study found that high birth weight (HBW) was related to an increased risk of heart failure[4]. Childhood body size was also reported to associate with a series of chronic conditions, including obesity, diabetes, metabolic syndrome, depression, and cancer[5–10]. However, evidence on whether and to what extent abnormal birth weight and childhood body size are associated with life-course risks of multiple chronic conditions, which is defined as multimorbidity[11], remains scanty.

Most previous studies mainly focused on a single measurement of birth weight or body size at a specific childhood phase, failing to assess the dynamic weight changes over time. An observational study combined five birth cohorts measured the weight gain in three age periods (0–2 years, 2 years to mid-childhood, and mid-childhood to adulthood) and found faster weight gain was associated with an increased risk of overweight and high blood pressure in adulthood[12]. Another cohort study reported an association between weight gain from birth to age 2 years and lung function in late life[13]. These findings endorsed the notion that childhood developmental differences are evident in the first few years after birth, and worsen during early childhood, resulting in continuous influence on health throughout life[14]. However, most current studies commonly measured weight gain or weight growth, ignoring the influence of shifting from a normal or plumper body size to a thinner body size. As an indicator of undernutrition or developmental delays, children with thinner body sizes were more likely to develop chronic conditions related to nutrition in later life[15]. Understanding the long-term consequences of different changing patterns of body size in early childhood is important for life-course monitoring and intervention of body weight and weight-related chronic diseases.

As birth weight and childhood body size are manageable[16–18], elucidating their life-course association with multimorbidity and chronic conditions would have implications for preventing health-related outcomes around the whole life-course. Leveraging data from 246,495 individuals from UK Biobank, our study aims to investigate the association of birth weight, childhood body size at 10 years, and their longitudinal changes with the late-life risk of 38 types of chronic conditions and multimorbidity. In our study, birth weight and childhood body size were associated with multimorbidity and many chronic conditions in later life. Individuals who changed body shape during childhood to be either thinner or plumper, no matter from which birth weight group, were more likely to develop more than one chronic disease later in life.

## Methods

**Study design and participants**. This cohort study used UK Biobank data and followed the Strengthening the Reporting of Observational Studies in Epidemiology (STROBE) reporting guideline. The UK Biobank is a large population-based prospective study, recruiting more than half a million UK individuals aged 40–69 years between 2006 and 2010. At baseline, sociodemographic, lifestyles, and health-related information were collected via touchscreen questionnaires and physical measurements. Participants were followed for health-related outcomes through linkage to routinely available national datasets, including the Hospital Episode Statistics for England, the Patient Episode Database for Wales, and the Scottish Morbidity Record for Scotland. Appropriate informed consent was obtained from participants, and ethical approval was covered by the UK Biobank. More details of the UK Biobank study are available elsewhere[19].

This study included individuals who attended the baseline assessment of UK Biobank. The exclusion criteria were: (1) individuals with no information on birth weight, childhood body size, or adulthood body mass index (BMI); (2) individuals who were part of a multiple birth; and (3) individuals who were lost or died during follow-up (Supplementary Fig. S1).

**Assessment of birth weight, childhood body size, and weight change**. The main exposures were birth weight and childhood body size at 10 years of each eligible participants, which were collected through self-reported questionnaires at baseline. Participants were asked to recall their birth weights and were categorized into three groups: LBW ( ≤ 2.5 kg), normal birth weight (NBW, 2.5–4.0 kg), and HBW (≥4.0 kg)[20,21]. Childhood body size was self-reported according to the question: "When you were 10 years old, compared to average would you describe yourself as…?", with the answer of being thinner, average, or plumper. Furthermore, we assessed the weight change from birth to childhood and classified weight change into 9 groups.

**Assessment of chronic conditions and multimorbidity**. Based on previous literature, including the Quality and Outcomes Framework (QoF, with the common conditions in the UK)[22], a large UK-based study[23], a systematic review on multimorbidity indices[24], and other evidence from UK Biobank on multimorbidity[23], a total of 38 physical and mental chronic conditions were included in our study to define multimorbidity (Supplementary Table S1). Information of aforementioned conditions was extracted through the linkage to inpatient hospital data, according to the codes of International Classification of Diseases 10th Revision (ICD-10). The onset time of each condition was recorded as the date of inpatient diagnosis. Multimorbidity was defined as the coexistence of two or more of these 38 chronic conditions.

**Covariates**. Covariates including sociodemographic characteristics (age, sex, ethnicity, Townsend deprivation index, and education levels), anthropometric information (BMI at baseline), lifestyle factors (current smoking and drinking status, physical activity, and intake of fruits and vegetables), and early life factors (maternal smoking status around birth and breastfeed status as a baby) were collected at baseline. Participants with missing data for covariates were assigned to a separate group of 'unknown'. Townsend deprivation index was divided into tertiles (low, moderate, and high). Adulthood BMI was categorized into 4 groups of underweight (<20.0 kg/m²), normal weight (20.0–24.9 kg/m²), overweight (25.0–29.9 kg/m²), and obesity (≥30.0 kg/m²)[25]. Physical activity was evaluated via Metabolic Equivalent Task (MET) minutes per week for moderate and vigorous activity and was further categorized into tertiles (low, moderate, and high). For the intake of fruits and vegetables, participants were categorized into four groups respectively, according to previous studies[26–28]: '<2.0 servings/day', '2.0–2.9 servings/day', '3.0–3.9 servings/day', and '≥4.0 servings/day'.

**Identification of multimorbidity patterns**. Exploratory factor analysis was performed with the principal factor method to identify multimorbidity patterns. The Kaiser-Meyer-Olkin

statistic, estimated to analyze the adequacy of the sample, was 0.74. The breakpoint of the screen plot, the eigenvalue (>1), and interpretability were used to determine multimorbidity patterns. We performed a varimax rotation of factor-loading matrices, with the resulting factor loadings representing the strength of the association between each chronic condition and the latent pattern. Chronic conditions with a factor loading > 0.35 referred to the characteristics of the multimorbidity pattern. Factor scores of all participants were calculated to indicate adherence to each multimorbidity pattern, as higher scores represented higher adherence. They were then divided into the percentage of 0–85%, 85–90%, 90–95%, and 95–100% to indicate adherence to each multimorbidity pattern for further analysis.

**Statistics and reproducibility**. Basic characteristics were described as mean (standard deviation [SD]) for continuous variables and as number (percentage) for categorical variables. The differences of variables across groups by birth weight and childhood body size were compared using one-way analysis of variance (ANOVA) and $\chi^2$ test.

In the primary analyses, Cox regression models were utilized to estimate the associations of birth weight and childhood body size with each chronic condition and multimorbidity. Years from birth to the onset of multimorbidity or each chronic condition, or 31 December 2020, whichever came first, were considered the time scale. The proportional hazard assumption was checked and verified using Schoenfeld's residual methods, where no violations were found. Four models were conducted to sequentially adjust covariates, including sociodemographic characteristics and anthropometric information, lifestyle factors, and early life factors. The restricted cubic spline (RCS) model was utilized to display the dose-response association of birth weight with multimorbidity and each chronic condition. Also, the cumulative incidence function (CIF) curves were used to compare the cumulative incidence of adulthood multimorbidity across different groups of birth weight and childhood body size. Moreover, associations of weight change with the incidence of each chronic condition and multimorbidity were assessed in each birth weight group using Cox proportional hazard models, with changes to average body size as reference groups. Besides, quasi-Poisson mixed effects models were used to evaluate the associations of birth weight, childhood body size, and weight change with the number of chronic conditions during follow-up.

A series of exploratory analyses, sensitivity analyses and subgroup analyses were conducted to test the robustness of our results. First, multinomial logistic regression models were used to explore the association of birth weight and childhood body size with different multimorbidity patterns. Second, we recategorized birth weight into four groups: ≤2.5 kg, 2.5-4.0 kg, 4.0-4.5 kg, and >4.5 kg[29,30], to explore the association of highest birth weight (>4.5 kg) with the risk of multimorbidity. Third, we repeated the primary analyses by: (1) excluding participants with missing data of covariates; (2) imputing missing data of covariates using multiple imputation for five times; and (3) including family history (mother/father/sibling) of several major diseases in the fully adjusted model: heart disease, stroke, cancer, chronic bronchitis/emphysema, high blood pressure, diabetes, Alzheimer's disease/dementia, Parkinson's disease, and severe depression (Supplementary Table S2). Fourth, subgroup analyses were performed stratified by age at baseline, sex, Townsend deprivation index, education level, and family history of major diseases. Potential effect modifications were evaluated by adding an interaction term of each variable and birth weight or childhood body size in the model.

Statistical analyses were performed using SAS (version 9.4, SAS Institute Inc., NC, USA) and R (version 4.1.2, R Foundation for Statistical Computing). Hazard ratios (HRs), odds ratios (ORs), relative risks (RRs) and 95% confidence interval (95% CI) were reported in this study. Significance tests were evaluated at the 0.05 level using two-sided tests.

**Ethical consideration**. The UK Biobank has ethical approval from the NHS National Research Ethics Service (16/NW/0274). This is a secondary analysis of UK Biobank which is exempted from Institutional Review Board.

**Reporting summary**. Further information on research design is available in the Nature Portfolio Reporting Summary linked to this article.

## Results

**Basic characteristics of participants**. Among the 502,490 participants attending baseline assessment of UK Biobank, a total of 246,495 participants remained in the current study according to the inclusion and exclusion criteria. The basic characteristics of excluded and included participants were summarized in Supplementary Table S3. Among the 246,495 participants (Supplementary Data 1), the mean baseline age was 55.2 (SD = 8.1) years, and 150,486 (61.1%) participants were female. The proportion of participants who reported their birth weight was 77.2% (NBW), 8.8% (LBW), and 14.0% (HBW), respectively. A total of 51.0% of participants reported an average childhood body size at 10 years, whereas 32.3% reported being slimmer and 16.6% reported being plumper. The characteristics of participants according to birth weight and body size at 10 years were also shown in Supplementary Data 1.

A total of 57,071 (23.2%) participants developed multimorbidity. The characteristics of participants by multimorbidity status were presented in Supplementary Data 2. Participants with multimorbidity were more likely to have abnormal birth weight (LBW or HBW) and childhood body size (thinner or plumper), higher BMI at baseline, and be physically inactive than those free of multimorbidity.

**Birth weight, childhood body size, and late-life multimorbidity and individual chronic conditions**. Participants with LBW/HBW and thinner/plumper childhood body size had a higher cumulative incidence of multimorbidity (Fig. 1). LBW (adjusted HR = 1.29, 95% CI = 1.26–1.33), HBW (adjusted HR = 1.02, 95% CI = > 1.00–1.05), thinner body size (adjusted HR = 1.21, 95% CI = 1.18–1.23), and plumper body size (adjusted HR = 1.06, 95% CI = 1.04–1.09) were all associated with a higher risk of developing multimorbidity (Table 1). The RCS model showed a U-shaped relationship between birth weight and multimorbidity (P for nonlinear <0.001, Fig. 2), with a lower risk at a birth weight of 3.33–4.09 kg. As shown in Table 2, low birth weight, high birth weight, thinner body size, and plumper body size were associated with increased number of chronic diseases.

Associations of birth weight and childhood body size with late-life individual chronic conditions were shown in Fig. 3. LBW and thinner body size was associated with higher risk of many chronic conditions. However, the results for HBW and plumper body size diverged. The U-shape relationship was also observed for birth weight with most of the 38 conditions (Supplementary Fig. S2–4).

**Weight change from birth to childhood and late life multimorbidity and individual chronic conditions**. Among participants with LBW, 43.2% kept being thinner at 10 years, and 15.4% became plumper. More than half of participants (51.6%) with

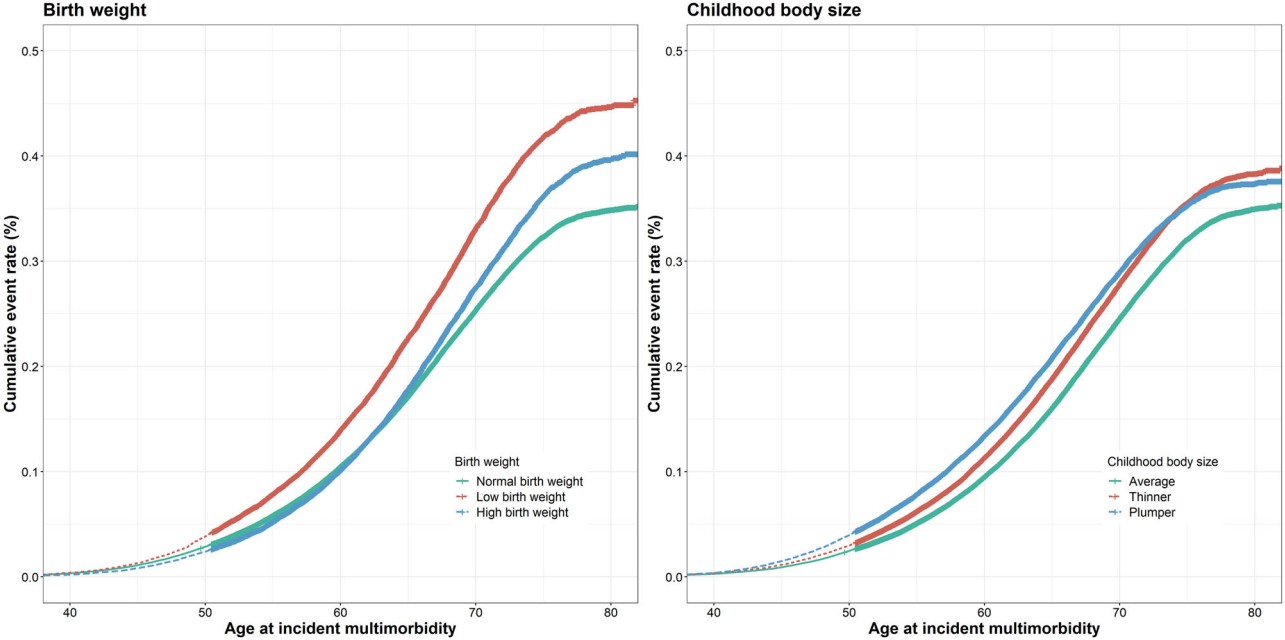

**Fig. 1 Cumulative incidence of multimorbidity according to birth weight and childhood body size.** For the curve of birth weight, the red, blue and green lines represent low birth weight, high birth weight, and normal birth weight, respectively. For the curve of childhood body size, the red, blue, and green lines represent thinner, plumper, and average body size, respectively.

NBW changed to be thinner at their 10 years. For participants with HBW, 26.1% of them were of average body size at 10 years, and 52.7% of them turned out to be thinner (Table 1). Changing to be thinner or plumper, no matter from which birth weight group, was observed to associate with higher risk of multimorbidity and increased number of chronic conditions, compared to changing to be average. The higher HRs occurred in changing to thinner body size (HR = 1.21, 95% CI = 1.15–1.28 for LBW to thinner body size; HR = 1.20, 95% CI = 1.18–1.23 for NBW to thinner body size; HR = 1.21, 95% CI = 1.15–1.27 for HBW to thinner body size). Also, changing to thinner or plumper body sizes were all associated with increased number of chronic conditions (Table 2).

The associations of weight change from birth to childhood with the risks of each chronic condition were showed in Fig. 4. For those with NBW, changing to be thinner at 10 years increased the risks of 28 conditions, whereas changing to be plumper was associated with higher risks of 12 conditions and lower risks of two conditions, respectively.

**Additional analyses.** According to factor analysis, we identified six patterns of multimorbidity (Supplementary Table S4). The associations of LBW and thinner body size with multimorbidity patterns were consistently observed, while plumper body size was associated with a lower risk of Cardiovascular Pattern (OR = 0.94, 95% CI = 0.89–0.99) (Supplementary Table S5). After recategorization of high birth weight, participants with the highest birth weight (>4.5 kg) had a higher risk of multimorbidity (adjusted HR = 1.10, 95% CI = 1.06–1.14), while that for participants with birth weight of 4.0-4.5 kg was not significant (Supplementary Table S6). Similar results were revealed after excluding or imputing the missing data of covariates (Supplementary Table S7-8). When additionally adjusting family history of several major disease in the model, similar results were found (Supplementary Table S9). The subgroup analyses showed similar results to our main findings (Supplementary Fig. S5). Moreover, the associations of birth weight and childhood body size with multimorbidity were consistent among participants with or

without family history of most major diseases (Supplementary Fig. S6).

## Discussion

In this large prospective cohort study of 246,495 participants, we found that birth weight and childhood body size were significantly associated with multimorbidity and many chronic conditions in late life. A U-shape relationship was observed between birth weight and multimorbidity and most chronic conditions. Individuals who changed to be thinner or plumper in childhood, no matter from which birth weight group, had increased risks of developing multimorbidity and many chronic conditions than those who changed to the average body size.

Previous cohort studies have explored the association between early life body size and several individual chronic conditions, mainly focusing on cardiometabolic diseases and cancers. Most of these studies have shown significant associations of birth weight and childhood body size with chronic conditions. For example, existing evidence has presented positive associations of LBW with higher risks of diabetes, metabolic syndrome, and cardiovascular disease (CVD)[3,31–33], and HBW with increased risks of obesity and schizophrenia[34,35]. Ahlgren et al. also found a positive linear relationship between birth weight and many types of cancers[36]. For body size in childhood, a higher BMI was observed to increase the risk of diabetes[37,38]. A systematic review and meta-analysis of 37 longitudinal studies also found that higher childhood BMI could increase the risk of diabetes, coronary heart disease and cancers, but not stroke or breast cancer[7]. Although the prevalence of multimorbidity has been increasing rapidly in recent decades, particularly among the aging population, whether it could be predicted by early life factors has not been well documented. Our study extends previous findings to a broader spectrum of physical and mental conditions to measure multimorbidity in a large study sample with long-term monitoring of health-related outcomes, showing associations of LBW, HBW, thinner and plumper body size with higher risks of 27, 11, 27, and 13 of the 38 chronic conditions respectively. Birth weight was an indicator of fetal health[2,39], and our results confirmed its

**Table 1 Associations of birth weight and childhood body size with multimorbidity.**

| Categories of birth weight, childhood body size, and weight change | | |
| --- | --- | --- |
| **Birth weight** | | |
| **Low birth weight** | **Normal birth weight** | **High birth weight** |
| *n* (%) | | |
| 21,626 (8.8) | 190,359 (77.2) | 34,510 (14.0) |
| Model 1 | 1.32 (1.28–1.35) | Reference | 1.05 (1.02–1.07) |
| Model 2 | 1.29 (1.26–1.33) | Reference | 1.02 (>1.00–1.05) |
| Model 3 | 1.29 (1.26–1.33) | Reference | 1.02 (>1.00–1.04) |
| Model 4 | 1.29 (1.26–1.33) | Reference | 1.02 (>1.00–1.05) |
| **Childhood body size** | | |
| **Thinner body size** | **Average body size** | **Plumper body size** |
| *n* (%) 79,700 (32.3) | 125,803 (51.0) | 40,992 (16.6) |
| Model 1 1.22 (1.20–1.24) | Reference | 1.05 (1.02–1.07) |
| Model 2 1.21 (1.18–1.23) | Reference | 1.06 (1.04–1.09) |
| Model 3 1.21 (1.18–1.23) | Reference | 1.06 (1.04–1.09) |
| Model 4 1.21 (1.18–1.23) | Reference | 1.06 (1.04–1.09) |
| **Groups of weight change** | | |
| **Low birth weight→Thinner** | **Low birth weight→Average** | **Low birth weight→Plumper** |
| *n* (%) 9,340 (43.2) | 8,950 (41.4) | 3,336 (15.4) |
| Model 1 1.24 (1.17–1.31) | Reference | 1.13 (1.06–1.22) |
| Model 2 1.21 (1.15–1.28) | Reference | 1.13 (1.05–1.21) |
| Model 3 1.21 (1.15–1.28) | Reference | 1.13 (1.05–1.21) |
| Model 4 1.21 (1.15–1.28) | Reference | 1.13 (1.05–1.22) |
| **Normal birth weight→Thinner** | **Normal birth weight→Average** | **Normal birth weight→Plumper** |
| *n* (%) 98,293 (51.6) | 61,740 (32.4) | 30,326 (15.9) |
| Model 1 1.22 (1.19–1.24) | Reference | 1.03 (1.00–1.06) |
| Model 2 1.20 (1.18–1.23) | Reference | 1.05 (1.02–1.08) |
| Model 3 1.20 (1.18–1.23) | Reference | 1.05 (1.02–1.07) |
| Model 4 1.20 (1.18–1.23) | Reference | 1.05 (1.02–1.07) |
| **High birth weight→Thinner** | **High birth weight→Average** | **High birth weight→Plumper** |
| *n* (%) 18,170 (52.7) | 9,010 (26.1) | 7,330 (21.2) |
| Model 1 1.24 (1.18–1.30) | Reference | 1.08 (1.02–1.14) |
| Model 2 1.21 (1.15–1.27) | Reference | 1.08 (1.03–1.14) |
| Model 3 1.21 (1.15–1.27) | Reference | 1.08 (1.03–1.14) |
| Model 4 1.21 (1.15–1.27) | Reference | 1.08 (1.03–1.14) |

Model 1: unadjusted.
Model 2: adjusted for age at baseline, sex, ethnicity, Townsend deprivation index, education levels, and BMI at baseline.
Model 3: model 2 also adjusted for current smoking and drinking status, physical activity, intake of fruits and vegetables.
Model 4: model 3 also adjusted for maternal smoking around birth and breastfed as a baby.
We adjusted for the other variable in each of the birth weight and childhood obesity status model.
Hazard ratios and 95% confidence intervals are shown in the table.

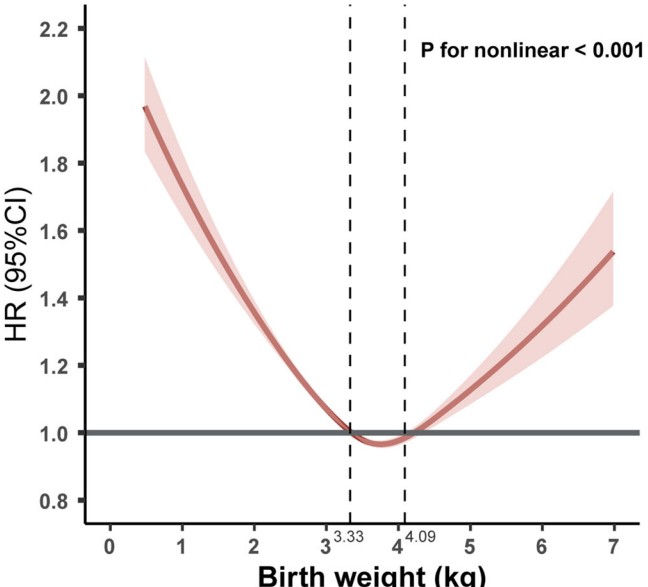

**Fig. 2 Restricted cubic spline models for the relationship of birth weight with and multimorbidity.** Adjusted for age at baseline, sex, ethnicity, Townsend deprivation index, education level, smoking status, drinking status, physical activity, intake of fruits and vegetables, body mass index categories at baseline, maternal smoking around birth and breastfed as a baby. The solid line represents hazard ratios, and the shading represents 95% confidence interval.

**Table 2 Association of birth weight, childhood body size, and weight change with the number of chronic conditions.**

| Categories of birth weight, childhood body size, and weight change | n (%) | RR (95% CI) |
| --- | --- | --- |
| **Birth weight** | | |
| Low birth weight | 21,626 (8.8) | 1.27 (1.25–1.29) |
| Normal birth weight | 190,359 (77.2) | Reference |
| High birth weight | 34,510 (14.0) | 1.05 (1.03–1.06) |
| **Childhood body size** | | |
| Thinner body size | 79,700 (32.3) | 1.18 (1.16–1.19) |
| Average body size | 125,803 (51.0) | Reference |
| Plumper body size | 40,992 (16.6) | 1.02 (1.01–1.03) |
| **Groups of weight change** | | |
| Low birth weight→Thinner | 9340 (43.2) | 1.18 (1.13–1.22) |
| Low birth weight→Average | 8950 (41.4) | Reference |
| Low birth weight→Plumper | 3336 (15.4) | 1.09 (1.03–1.14) |
| Normal birth weight→Thinner | 98,293 (51.6) | 1.17 (1.15–1.19) |
| Normal birth weight→Average | 61,740 (32.4) | Reference |
| Normal birth weight→Plumper | 30,326 (15.9) | 1.01 (0.99–1.02) |
| High birth weight→Thinner | 18,170 (52.7) | 1.20 (1.16–1.24) |
| High birth weight→Average | 9010 (26.1) | Reference |
| High birth weight→Plumper | 7330 (21.2) | 1.04 (1.01–1.08) |

Adjusted for age at baseline, sex, ethnicity, Townsend deprivation index, education level, smoking status, drinking status, physical activity, intake of fruits and vegetables, BMI categories at baseline, maternal smoking around birth and breastfed as a baby.
Relative risks and 95% confidence intervals are shown in the table.

association with multimorbidity and many chronic conditions in late life, especially LBW. In addition, the consideration of childhood body size further expanded the exploration of early life risk factors to a 10-year range. Our findings can support the "fetal origins" and DOHaD hypotheses that the disproportionate growth in early life, in utero in particular, would determine the development of many chronic diseases in late life[40], with substantial implications for individuals, families, societies, and the global healthcare systems.

It is worth noting that individuals with HBW were found to experience lower risks of asthma, diabetes, hypertension, and endometriosis, according to our results. The RCS models found the U-shape dose-response relationships between birth weight and three of these conditions (asthma, diabetes, and hypertension), and the lower HRs occurred in higher birth weight (>4.0 kg) than those for other conditions, which might explain the inverse association of HBW with these three conditions. We further conducted a sensitivity analysis by recategorizing HBW

into 4.0–4.5 kg and >4.5 kg groups and found the significant association with multimorbidity disappeared in the 4.0–4.5 kg group, while that for the highest birth weight (>4.5 kg) group remained. These results suggested that highest birth weight may be a better indicator of multimorbidity and several diseases, which might also partially explain the inconsistency with previous evidence (e.g., a previous study suggested the higher risk of

| | Birth weight | | Childhood body size | |
| --- | --- | --- | --- | --- |
| | Low birth weight | High birth weight | Thinner | Plumper |
| Atrial fibrillation | 0.95 (0.77-1.16) | 1.01 (0.87-1.17) | 1.10 (0.97-1.25) | 1.20 (1.02-1.40) |
| Angina | 1.38 (1.29-1.47) | 1.19 (1.12-1.26) | 1.27 (1.21-1.33) | 0.95 (0.90-1.01) |
| Anxiety | 1.24 (1.14-1.36) | 1.00 (0.93-1.08) | 1.20 (1.13-1.27) | 1.07 (0.99-1.15) |
| Asthma | 1.31 (1.25-1.37) | 0.93 (0.89-0.97) | 1.33 (1.28-1.37) | 1.14 (1.10-1.19) |
| Bronchiectasis | 1.23 (1.02-1.50) | 1.15 (0.97-1.35) | 1.23 (1.08-1.40) | 0.93 (0.78-1.11) |
| Cancer | 1.05 (1.00-1.09) | 1.11 (1.07-1.15) | 1.01 (0.98-1.03) | 0.91 (0.88-0.94) |
| Cirrhosis | 1.75 (1.30-2.35) | 1.04 (0.77-1.40) | 1.00 (0.79-1.27) | 0.93 (0.71-1.23) |
| Chronic kidney disease | 1.42 (1.27-1.58) | 0.97 (0.88-1.08) | 1.16 (1.07-1.26) | 0.97 (0.88-1.08) |
| Chronic obstructive pulmonary disease | 1.68 (1.53-1.84) | 1.18 (1.08-1.29) | 1.26 (1.18-1.35) | 1.12 (1.03-1.23) |
| Dementia | 1.28 (0.90-1.81) | 1.02 (0.73-1.42) | 1.17 (0.91-1.51) | 1.12 (0.81-1.54) |
| Depression | 1.24 (1.15-1.33) | 0.95 (0.89-1.01) | 1.25 (1.18-1.31) | 1.25 (1.17-1.32) |
| Diabetes | 1.49 (1.41-1.58) | 0.94 (0.89-0.99) | 1.38 (1.33-1.44) | 1.12 (1.06-1.17) |
| Eczema or dermatitis | 1.04 (0.90-1.20) | 1.08 (0.96-1.21) | 1.20 (1.09-1.31) | 1.01 (0.90-1.14) |
| Epilepsy | 1.34 (1.16-1.54) | 1.11 (0.98-1.26) | 1.09 (0.99-1.21) | 0.96 (0.85-1.09) |
| Glaucoma | 1.30 (1.16-1.45) | 1.13 (1.02-1.24) | 1.03 (0.95-1.11) | 0.99 (0.90-1.10) |
| Hepatitis | 2.16 (1.11-4.18) | 1.76 (0.98-3.17) | 0.96 (0.54-1.70) | 1.59 (0.89-2.84) |
| Heart failure | 1.30 (1.14-1.47) | 1.21 (1.09-1.35) | 1.21 (1.10-1.32) | 1.04 (0.93-1.16) |
| Hypertension | 1.32 (1.28-1.36) | 0.96 (0.93-0.98) | 1.19 (1.17-1.22) | 1.04 (1.01-1.07) |
| Irritable bowel syndrome | 1.35 (1.23-1.48) | 0.94 (0.86-1.04) | 1.28 (1.20-1.37) | 1.11 (1.02-1.21) |
| Inflammatory bowel disease | 1.06 (0.93-1.20) | 1.11 (1.00-1.23) | 1.15 (1.06-1.25) | 0.96 (0.87-1.08) |
| Myocardial infarction | 1.21 (1.09-1.34) | 1.14 (1.04-1.24) | 1.17 (1.09-1.25) | 0.89 (0.81-0.97) |
| Migraine | 1.07 (0.93-1.23) | 0.96 (0.85-1.08) | 1.23 (1.12-1.35) | 1.08 (0.96-1.21) |
| Multiple sclerosis | 1.25 (1.00-1.57) | 0.99 (0.81-1.21) | 0.87 (0.74-1.02) | 1.34 (1.12-1.60) |
| Osteoporosis | 1.47 (1.31-1.64) | 1.09 (0.97-1.22) | 1.05 (0.97-1.15) | 1.09 (0.97-1.22) |
| Parkinson's disease | 1.14 (0.84-1.54) | 1.29 (1.02-1.63) | 1.11 (0.91-1.35) | 0.91 (0.70-1.18) |
| Prostate problem | 0.91 (0.83-1.01) | 1.45 (1.36-1.55) | 1.16 (1.09-1.23) | 0.82 (0.76-0.89) |
| Peripheral vascular disease | 1.49 (1.30-1.71) | 1.05 (0.92-1.20) | 1.03 (0.92-1.14) | 1.32 (1.17-1.49) |
| Rheumatoid arthritis | 1.11 (0.95-1.30) | 0.96 (0.83-1.10) | 1.20 (1.08-1.34) | 1.06 (0.93-1.22) |
| Schizophrenia | 1.25 (0.85-1.83) | 1.06 (0.75-1.49) | 1.73 (1.32-2.26) | 1.28 (0.91-1.79) |
| Chronic sinusitis | 1.17 (1.00-1.38) | 1.06 (0.92-1.22) | 1.21 (1.08-1.35) | 1.08 (0.94-1.24) |
| Stroke | 1.30 (1.13-1.50) | 1.05 (0.92-1.19) | 1.23 (1.11-1.35) | 0.99 (0.88-1.13) |
| Thyroid problem | 1.37 (1.30-1.45) | 0.95 (0.90-1.00) | 1.10 (1.06-1.15) | 1.10 (1.04-1.15) |
| Dyspepsia | 1.30 (1.21-1.39) | 1.01 (0.95-1.08) | 1.12 (1.06-1.18) | 1.05 (0.99-1.12) |
| Constipation | 1.28 (1.19-1.39) | 1.07 (0.99-1.14) | 1.17 (1.11-1.24) | 1.11 (1.04-1.19) |
| Hearing loss | 1.38 (1.22-1.56) | 1.26 (1.13-1.40) | 1.15 (1.05-1.25) | 0.95 (0.85-1.07) |
| Diverticular disease of intestine | 1.12 (1.06-1.17) | 1.04 (0.99-1.08) | 1.19 (1.15-1.23) | 1.00 (0.96-1.04) |
| Endometriosis | 1.19 (1.06-1.34) | 0.72 (0.64-0.81) | 1.19 (1.10-1.28) | 1.29 (1.17-1.42) |
| Meniere's disease | 1.25 (0.91-1.72) | 1.37 (1.06-1.77) | 1.43 (1.15-1.78) | 1.17 (0.89-1.53) |

**Fig. 3 Estimated hazard ratios (95% confidence intervals) for the association of birth weight and childhood body size with incident individual conditions.** Reference groups: normal birth weight and average body size in childhood. Adjusted for age at baseline, sex, ethnicity, Townsend deprivation index, education level, smoking status, drinking status, physical activity, intake of fruits and vegetables, body mass index categories at baseline, maternal smoking around birth and breastfed as a baby. Diseases with hazard ratios>1 and $P < 0.05$ were marked in red, those with hazard ratios<1 and $P < 0.05$ were marked in blue.

diabetes in those with HBW[41]). Besides, the negative linear relationship between birth weight and endometriosis has not been reported previously, calling for further clarification of such an association. In addition, the inverse association of plumper body size at 10 years with cancer, myocardial infarction, and prostate problems were observed in our study. Also, the inverse association of plumper body size with myocardial infarction might contribute to the inverse association between plumper body size and Cardiovascular Pattern. Consistent with our findings, the association of plumper body size in childhood could diverge with different subtype cancers, according to a previous study[42]. These discrepancies could partly be explained by the "growth acceleration" hypothesis that the catch-up growth for individuals with LBW or thinner childhood body size would result in relatively higher postnatal or pubertal growth and consequently lead to a higher risk of several conditions than those with HBW or plumper body size[43]. However, due to the lack of continuous data on childhood body size, we could not find the cut-off of childhood BMI on multimorbidity and chronic conditions. Further

large population-based studies are warranted to validate the role of HBW and plumper body size in childhood on the observed associations.

A previous study of UK Biobank observed a significant interaction between birth weight and adult BMI[3], indicating that the longitudinal weight change might relate to long-term health status. Therefore, we further investigated the life-course association of weight change from birth to childhood with multimorbidity and chronic conditions. We found the association of changing to be thinner or plumper with multimorbidity was robust, regardless of individuals' birth weight. In line with our findings, a systematic review reported that individuals changing from normal weight to high weight in adulthood would experience higher CVD risk factors and outcomes[44], supporting that unfavorable weight change may adversely impact subsequent health-related outcomes. However, no previous study has well explored the effect of weight change from birth to childhood. As a proxy indicator of early life growth, more attention should be paid to exploring the association of weight change from birth to

| | Low birth weight | | Normal birth weight | | High birth weight | |
| --- | --- | --- | --- | --- | --- | --- |
| | Thinner | Plumper | Thinner | Plumper | Thinner | Plumper |
| Atrial fibrillation | 1.07 (0.58-1.99) | 1.53 (0.71-3.30) | 1.14 (0.98-1.33) | 1.26 (1.05-1.52) | 0.99 (0.70-1.38) | 1.06 (0.72-1.57) |
| Angina | 1.30 (1.14-1.48) | 0.99 (0.83-1.18) | 1.28 (1.21-1.35) | 0.94 (0.88-1.01) | 1.17 (1.04-1.32) | 0.96 (0.84-1.09) |
| Anxiety | 1.24 (1.04-1.48) | 1.07 (0.84-1.36) | 1.21 (1.13-1.30) | 1.04 (0.95-1.14) | 1.07 (0.90-1.27) | 1.17 (0.98-1.40) |
| Asthma | 1.29 (1.17-1.43) | 1.22 (1.07-1.38) | 1.31 (1.27-1.37) | 1.12 (1.06-1.17) | 1.43 (1.30-1.58) | 1.22 (1.10-1.35) |
| Bronchiectasis | 1.07 (0.72-1.60) | 1.01 (0.58-1.75) | 1.22 (1.05-1.42) | 0.91 (0.73-1.12) | 1.47 (1.04-2.07) | 0.96 (0.62-1.50) |
| Cancer | 1.08 (0.99-1.18) | 0.95 (0.84-1.08) | 0.99 (0.96-1.03) | 0.90 (0.86-0.94) | 1.02 (0.94-1.10) | 0.93 (0.86-1.01) |
| Cirrhosis | 0.58 (0.31-1.06) | 0.41 (0.17-0.99) | 1.11 (0.84-1.47) | 1.07 (0.77-1.48) | 1.09 (0.57-2.09) | 1.00 (0.50-1.99) |
| Chronic kidney disease | 1.20 (0.96-1.51) | 1.23 (0.94-1.63) | 1.12 (1.02-1.23) | 0.92 (0.81-1.04) | 1.44 (1.14-1.81) | 1.01 (0.79-1.30) |
| Chronic obstructive pulmonary disease | 1.20 (1.00-1.43) | 1.07 (0.84-1.37) | 1.27 (1.16-1.38) | 1.13 (1.02-1.25) | 1.32 (1.10-1.60) | 1.12 (0.91-1.37) |
| Dementia | 1.94 (0.96-3.89) | 0.45 (0.12-1.65) | 0.93 (0.69-1.26) | 1.17 (0.81-1.69) | 1.81 (0.84-3.88) | 1.62 (0.68-3.82) |
| Depression | 1.17 (1.01-1.36) | 1.47 (1.23-1.76) | 1.26 (1.19-1.33) | 1.23 (1.15-1.31) | 1.25 (1.09-1.44) | 1.20 (1.03-1.40) |
| Diabetes | 1.33 (1.19-1.50) | 1.23 (1.07-1.41) | 1.39 (1.32-1.46) | 1.08 (1.02-1.15) | 1.41 (1.24-1.59) | 1.19 (1.06-1.35) |
| Eczema or dermatitis | 1.09 (0.80-1.47) | 1.17 (0.79-1.73) | 1.23 (1.10-1.36) | 1.00 (0.87-1.15) | 1.11 (0.86-1.43) | 0.99 (0.75-1.31) |
| Epilepsy | 1.14 (0.85-1.53) | 1.36 (0.95-1.96) | 1.12 (1.00-1.26) | 0.96 (0.83-1.12) | 0.95 (0.73-1.25) | 0.77 (0.56-1.04) |
| Glaucoma | 0.87 (0.69-1.10) | 1.06 (0.78-1.44) | 1.03 (0.94-1.13) | 1.02 (0.90-1.15) | 1.21 (0.99-1.48) | 0.85 (0.67-1.09) |
| Hepatitis | 0.45 (0.09-2.34) | 1.75 (0.45-6.75) | 0.69 (0.32-1.49) | 1.72 (0.86-3.47) | 3.14 (1.02-9.68) | 0.95 (0.18-5.02) |
| Heart failure | 0.99 (0.75-1.31) | 1.25 (0.90-1.72) | 1.28 (1.15-1.43) | 1.01 (0.88-1.16) | 0.99 (0.78-1.26) | 1.00 (0.78-1.27) |
| Hypertension | 1.16 (1.09-1.23) | 1.09 (1.01-1.18) | 1.19 (1.16-1.22) | 1.02 (0.99-1.05) | 1.24 (1.17-1.31) | 1.09 (1.03-1.16) |
| Irritable bowel syndrome | 1.31 (1.08-1.59) | 1.18 (0.90-1.53) | 1.30 (1.20-1.40) | 1.12 (1.01-1.24) | 1.18 (0.97-1.44) | 1.03 (0.83-1.29) |
| Inflammatory bowel disease | 1.07 (0.82-1.39) | 1.07 (0.74-1.53) | 1.18 (1.08-1.30) | 0.98 (0.86-1.12) | 1.05 (0.84-1.31) | 0.84 (0.65-1.09) |
| Myocardial infarction | 1.14 (0.93-1.40) | 0.87 (0.64-1.17) | 1.18 (1.08-1.27) | 0.90 (0.81-1.00) | 1.13 (0.94-1.35) | 0.84 (0.69-1.04) |
| Migraine | 1.38 (1.04-1.83) | 0.89 (0.59-1.36) | 1.22 (1.10-1.35) | 1.15 (1.01-1.31) | 1.18 (0.91-1.52) | 0.85 (0.63-1.15) |
| Multiple sclerosis | 1.25 (0.78-2.02) | 1.57 (0.86-2.86) | 0.83 (0.69-1.00) | 1.23 (1.00-1.53) | 0.78 (0.47-1.28) | 1.78 (1.16-2.74) |
| Osteoporosis | 1.13 (0.89-1.42) | 1.23 (0.89-1.69) | 1.05 (0.95-1.16) | 1.01 (0.88-1.16) | 1.01 (0.79-1.30) | 1.32 (1.02-1.71) |
| Parkinson's disease | 1.22 (0.64-2.34) | 1.02 (0.42-2.50) | 1.07 (0.85-1.34) | 0.84 (0.61-1.15) | 1.17 (0.71-1.94) | 1.03 (0.59-1.80) |
| Prostate problem | 1.08 (0.89-1.32) | 0.63 (0.46-0.87) | 1.16 (1.09-1.24) | 0.82 (0.75-0.90) | 1.17 (1.03-1.34) | 0.89 (0.76-1.04) |
| Peripheral vascular disease | 0.95 (0.71-1.26) | 1.06 (0.73-1.53) | 1.01 (0.89-1.14) | 1.34 (1.17-1.55) | 1.18 (0.88-1.59) | 1.41 (1.04-1.89) |
| Rheumatoid arthritis | 1.24 (0.89-1.72) | 0.91 (0.56-1.45) | 1.19 (1.06-1.35) | 1.02 (0.87-1.20) | 1.17 (0.86-1.61) | 1.34 (0.98-1.85) |
| Schizophrenia | 1.43 (0.64-3.21) | 1.11 (0.41-3.02) | 1.74 (1.27-2.37) | 1.37 (0.92-2.03) | 1.95 (0.96-3.95) | 1.07 (0.45-2.54) |
| Chronic sinusitis | 1.04 (0.74-1.45) | 1.16 (0.74-1.81) | 1.27 (1.12-1.44) | 1.05 (0.89-1.24) | 1.02 (0.75-1.40) | 1.12 (0.81-1.54) |
| Stroke | 1.29 (0.96-1.72) | 1.23 (0.84-1.81) | 1.25 (1.11-1.40) | 0.94 (0.81-1.09) | 1.03 (0.78-1.37) | 1.05 (0.78-1.40) |
| Thyroid problem | 1.26 (1.12-1.42) | 1.25 (1.08-1.45) | 1.07 (1.02-1.12) | 1.07 (1.01-1.13) | 1.15 (1.02-1.29) | 1.13 (1.00-1.27) |
| Dyspepsia | 1.15 (0.99-1.34) | 1.29 (1.06-1.56) | 1.10 (1.04-1.16) | 1.04 (0.96-1.12) | 1.25 (1.09-1.43) | 1.01 (0.86-1.18) |
| Constipation | 1.18 (1.00-1.39) | 1.13 (0.91-1.41) | 1.16 (1.09-1.24) | 1.09 (1.01-1.19) | 1.20 (1.03-1.40) | 1.18 (1.00-1.40) |
| Hearing loss | 1.11 (0.86-1.43) | 1.05 (0.75-1.48) | 1.12 (1.01-1.24) | 0.94 (0.82-1.08) | 1.29 (1.04-1.61) | 0.94 (0.73-1.21) |
| Diverticular disease of intestine | 1.12 (1.01-1.24) | 1.00 (0.87-1.14) | 1.20 (1.15-1.24) | 1.00 (0.96-1.05) | 1.21 (1.11-1.32) | 0.99 (0.90-1.09) |
| Endometriosis | 1.26 (1.00-1.60) | 1.14 (0.83-1.57) | 1.20 (1.10-1.31) | 1.30 (1.17-1.45) | 0.95 (0.73-1.25) | 1.31 (1.00-1.71) |
| Meniere's disease | 1.32 (0.65-2.68) | 2.49 (1.13-5.47) | 1.51 (1.16-1.95) | 1.07 (0.76-1.49) | 1.25 (0.73-2.15) | 1.05 (0.59-1.89) |

**Fig. 4 Association of weight change from birth to childhood with individual conditions.** Reference groups: changing to average childhood body size. Adjusted for age at baseline, sex, ethnicity, Townsend deprivation index, education level, smoking status, drinking status, physical activity, intake of fruits and vegetables, body mass index categories at baseline, maternal smoking around birth and breastfed as a baby. Diseases with hazard ratios >1 and $P < 0.05$ were marked in red, those with hazard ratios <1 and $P < 0.05$ were marked in blue.

childhood and its trajectories with the development of multi-morbidity and chronic conditions.

When considering family history of several chronic diseases in the model, the associations of birth weight and childhood body size with multimorbidity were similar to the main results. Also, such association was consistent among participants with or without family history of most diseases, implying that the association of birth weight and childhood body size with multimorbidity may be independent of familial genetic factors. According to a Mendelian randomization study which provided evidence of genetic pleiotropy between birth weight and cardiometabolic risk factors, maternal intrauterine environment, in which maternal genetic variation influences offspring birth weight, was not the major predictor of offspring cardiometabolic outcomes, but birth weight determined by the individuals' own genetic factors was more likely to associate with cardiometabolic outcomes[45]. This study, coupled with our results, provided converging evidence that familial genetic effects that predispose to offspring birth weight, were not the main determinant of some chronic diseases and multimorbidity in an individual's later life. Future studies using genetic data are warranted to verify such points.

The main strengths of this study include the large sample size, the long-term follow-up, the measurement of weight change, and the consideration of wide-range chronic conditions to measure multimorbidity using longitudinal data from UK Biobank. Limitations to this study also warrant consideration. First, the information on birth weight and childhood body size was self-reported by individuals, which was vulnerable to recall bias. Also, the childhood body size had no clear definition and was subjectively judged by individuals themselves, leading to the possibility of misclassification. Second, nearly half of participants recruited at baseline were excluded in our study, whose characteristics differed from the included, limiting the generalizability of our results. Third, the information on chronic conditions was linked to inpatient hospital data, missing out on the mild or undiagnosed forms of conditions, like depression and migraine. Fourth, the association of interest covered a large time span from birth to late-life, despite the inclusion of multiple covariates in our study, the potential for residual confounding of other factors could not been excluded. Fifth, although gestational age is an important factor affecting birth weight, we are not available to this data in UK Biobank. Last, we used family history of several major chronic disease as a proxy for genetic factors, future research utilizing genetic data are warranted to explore the interaction between genetic factors with birth weight and childhood body size in predicting multimorbidity.

In conclusion, birth weight, childhood body size and their changes are associated with higher risks of developing multimorbidity and many chronic diseases in late life. Early

monitoring and interventions to keep a normal body size in childhood could have life-course benefits for preventing multi-morbidity above and beyond individual chronic conditions.

## Data availability

This research was conducted using the UK Biobank study under Application Number 66354. The original data for the study are available on the website: https://www.ukbiobank.ac.uk/. Detailed data on the demographics of study participants are shown in Supplementary Data 1 and Supplementary Data 2. All other data are available from the corresponding author (or other sources, as applicable) on reasonable request.

## Code availability

The analytic code was shown in Supplementary Data 3.

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

## Acknowledgements

This work was funded by Zhejiang University. We thank all the lab members who gave support to the study and all the researchers and staff involved in UK Biobank.

## Author contributions

All authors contributed to the study conception and design. X.X. contributed to the study conceptualization and supervised the whole project. Yu.Z. made the analysis plan, conducted the statistical analyses, and drafted and revised the manuscript. Ya.Z. verified the underlying data and drafted the initial manuscript. Y.C., R.M.C.-L., M.F., and S.C. provided support on the statistical methods and manuscript revision. All authors contributed to and approved the final manuscript. X.X. is the corresponding authors, had full access to all the data, and had final responsibility for the decision to submit for publication.

## Competing interests

The authors declare no competing interests.
