## [Peer Review File · Communications Medicine]

Reviewers' comments:

Reviewer #1 (Remarks to the Author):

MAIN COMMENT

A prospective cohort study of 246,495 UK Biobank participants (aged 40-69 years) who reported birth weight and childhood body size at ten years old examined the relationships between birth weight categorized into low, normal, and high, childhood body size was reported as being thinner, average, or plumper, and multimorbidity defined as having two or more of 38 chronic conditions retrieved from inpatient hospital data. Low birth weight, high birth weight, thinner body size, and plumper body size were associated with higher risks of multimorbidity. A U-shaped relationship between birth weight and multimorbidity was observed.

The manuscript is well written and organized, the tables and figures are appropriate, the inferences and discussion are thoughtful, and the limitations are adequately acknowledged. Modeling death as a competing risk is of interest in place of excluding people who died, but this is probably beyond the authors' original aim. Otherwise, my only request is for minor grammar edits to avoid referring to associations as effects.

Overall, the authors submitted a very nice manuscript that adds to existing literature.

Specific Comments

1. Throughout: please refer to associations rather than effects.
2. Line 74. Delete 'preventable'.
3. Line 91-96. Typically, the numeric results for sample derivation are described in a result section, while the underlying rationale for exclusions is provided in a methods section.
4. Line 183. With rounding of the lower confidence limit, it appears the association is not statistically significant—please clarify using a table note or otherwise (e.g., CI= >1.00-1.05).
5. Line 215. Why are odds ratios used in place of hazard ratios?
6. Line 215. Please provide the sample size for each derived multimorbidity pattern group in Table S4.

Reviewer #2 (Remarks to the Author):

This is an observational study based on self-reported data on birth weight and body shape at age 10 years in a very large group of participants in the UK biobank study. The authors have used the huge material to conduct a series of statistical analyses. The finding in general is that low birth weight and thinness in childhood are associated with worse outcomes according to life-long accumulation of multi-morbidities based on hospital records. The methods and findings seem to be robust, and adjustment has also been carried out for social factors including the Townsend deprivation index. One obvious limitation, besides the self-report of birth weight, is the lack of data on gestational age, only possible to get access to via midwife reports, or national registers. This should be acknowledged. The role of genetics could be touched upon as well (the nature-nurture controversy). Is, for example, anything known about family history of major disease categories (CVD, DM, cancer) as a proxy for genetic influences? Could adjustment or stratification for family history bring a further insight?

Reviewer 1:

A prospective cohort study of 246,495 UK Biobank participants (aged 40-69 years) who reported birth weight and childhood body size at ten years old examined the relationships between birth weight categorized into low, normal, and high, childhood body size was reported as being thinner, average, or plumper, and multimorbidity defined as having two or more of 38 chronic conditions retrieved from inpatient hospital data. Low birth weight, high birth weight, thinner body size, and plumper body size were associated with higher risks of multimorbidity. A U-shaped relationship between birth weight and multimorbidity was observed.

The manuscript is well written and organized, the tables and figures are appropriate, the inferences and discussion are thoughtful, and the limitations are adequately acknowledged. Modeling death as a competing risk is of interest in place of excluding people who died, but this is probably beyond the authors' original aim. Otherwise, my only request is for minor grammar edits to avoid referring to associations as effects. Overall, the authors submitted a very nice manuscript that adds to existing literature.

Response: We would like to thank the reviewer's positive remarks on the manuscript. We have revised the manuscript to avoid referring to associations as effects.

Specific Comments

1. Throughout: please refer to associations rather than effects.

Response: We have corrected this throughout the manuscript.

2. Line 74. Delete 'preventable'.

Response: We have deleted 'preventable' in this sentence.

3. Line 91-96. Typically, the numeric results for sample derivation are described in a result section, while the underlying rationale for exclusions is provided in a methods section.

Response: We have rephrased this part in Methods and Results to make it clear:

Methods:

“This study included individuals who attended the baseline assessment of UK Biobank. The exclusion criteria were: (1) individuals with no information on birth weight, childhood body size, or adulthood body mass index (BMI); (2) individuals who were part of a multiple birth; and (3) individuals who were lost or died during follow-up (**Supplementary Figure S1**).” (page 5, line 91-94)

Results:

“Among the 502,490 participants attending baseline assessment of UK Biobank, a total of 246,495 participants remained in the current study according to the inclusion and exclusion criteria. The basic characteristics of excluded and included participants were summarized in **Supplementary Table S2**.” (page 8, line 167-170)

4. Line 183. With rounding of the lower confidence limit, it appears the association is not statistically significant—please clarify using a table note or otherwise (e.g., CI= >1.00-1.05).

Response: We have revised the 95% CI as >1.00-1.05.

5. Line 215. Why are odds ratios used in place of hazard ratios?

Response: We used exploratory factor analysis to identify multimorbidity patterns through which each participant would obtain a factor score in each multimorbidity pattern. We then divided factor scores into the percentage of 0-85%, 85%-90%, 90%-95%, and 95%-100% to each multimorbidity pattern as a higher factor score indicating higher adherence to the pattern. The outcome variables of each multimorbidity pattern were multinomial variables, hence multinomial logistic regression models were used. Odds ratios of the highest group of factor scores compared with the lowest group for each multimorbidity pattern were reported:

“First, multinomial logistic regression models were used to explore the association of birth weight and childhood body size with different multimorbidity patterns (identified by exploratory factor analysis, see **Supplementary method**).” (page 7, line 150-152)

6. Line 215. Please provide the sample size for each derived multimorbidity pattern group in Table S4.

Response: As mentioned above, each participant would obtain a factor score in each multimorbidity pattern and factor scores were divided into four groups as the percentage of 0-85%, 85%-90%, 90%-95%, and 95%-100%. Therefore, there is no actual sample size for each multimorbidity pattern.

Reviewer 2:

This is an observational study based on self-reported data on birth weight and body shape at age 10 years in a very large group of participants in the UK biobank study. The authors have used the huge material to conduct a series of statistical analyses. The finding in general is that low birth weight and thinness in childhood are associated with worse outcomes according to life-long accumulation of multimorbidities based on hospital records. The methods and findings seem to be robust, and adjustment has also been carried out for social factors including the Townsend deprivation index.

One obvious limitation, besides the self-report of birth weight, is the lack of data on gestational age, only possible to get access to via midwife reports, or national registers. This should be acknowledged. The role of genetics could be touched upon as well (the nature-nurture controversy). Is, for example, anything known about family history of major disease categories (CVD, DM, cancer) as a proxy for genetic influences? Could adjustment or stratification for family history bring a further insight?

Response: We really appreciate your valuable comments which is important to improve the manuscript. First, we have acknowledged the lack of data on gestational age in this study in limitation:

“Fifth, although gestational age is an important factor affecting birth weight, we are not available to this data in UK Biobank.” (page 14, line 322-323)

Second, we acknowledge the role of genetics in the association between birth weight and some chronic conditions and have added corresponding descriptions in Discussion. However, we could not explore the genetic factors in such association since we are not available to data on family history of chronic conditions. This limitation has been added to the manuscript.

Discussion:

“The role of genetics could also explain our results. Some chronic diseases-related loci are found to directly associated with birth weight^{1,2}, for example, a population-based study combining two nested case-control studies found interaction between birth weight and obesity genotype score in predicting the risk of diabetes, and genetic effects were more apparent in individuals with LBW than in those with HBW³. These findings may explain our result that LBW is associated with a higher risk of diabetes than HBW. Another Mendelian randomization study provided evidence of genetic pleiotropy between birth weight and cardiometabolic risk factors, suggesting that maternal intrauterine environment, in which maternal genetic variation influences offspring birth weight, was not the major predictor of offspring cardiometabolic outcomes, but birth weight determined by the individuals’ own genetic factors was more likely to associate with cardiometabolic outcomes⁴. These findings indicate that the role of genetic factors on the association between early-life body weight and multimorbidity should be highlighted. However, we did not consider genetic factors in this study due to the lack of data on family history of chronic diseases,

future studies are warranted to explore the genetic role in such associations.”
(page 13, line 294-308)

Limitation:

“Last, the interaction between genetics and birth weight in predicting chronic conditions was previously reported, but we could not take genetic factors into consideration due to the lack of data on family history of chronic conditions.”

(page 14, line 324-326)

References

1. Freathy RM, Weedon MN, Bennett A *et al.* Type 2 diabetes TCF7L2 risk genotypes alter birth weight: a study of 24,053 individuals. *Am J Hum Genet* 2007;**80**:1150-61.
2. Freathy RM, Bennett AJ, Ring SM *et al.* Type 2 Diabetes Risk Alleles Are Associated With Reduced Size at Birth. *Diabetes* 2009;**58**:1428-33.
3. Li Y, Qi Q, Workalemahu T, Hu FB, Qi L. Birth weight, genetic susceptibility, and adulthood risk of type 2 diabetes. *Diabetes Care* 2012;**35**:2479-84.
4. Moen GH, Brumpton B, Willer C *et al.* Mendelian randomization study of maternal influences on birthweight and future cardiometabolic risk in the HUNT cohort. *Nat Commun* 2020;**11**:5404.

Reviewers' comments:

Reviewer #1 (Remarks to the Author):

Thank you for responding to the previous critique, I have no further comments.

Best wishes,
Steven Korzeniewski

Reviewer #2 (Remarks to the Author):

The manuscript has improved following substantial revision and adding text on the role of genetic factors for observed associations, as well as lack of data on gestational age in the UK Biobank. What should be further revised is that you mix up lack of data on genetics (that is available in UK Biobank, see many previous publications) with lack of data on family history (that you mention is not available) which covers a wider range of risk markers/traits/habits than only genetics (the so called "missing heritability"). However, I found evidence for that family history is in fact available in the UK Biobank questionnaire dataset, as in most other cohort studies, see link: <https://biobank.ndph.ox.ac.uk/ukb/label.cgi?id=100034>

This has also been documented in some publications, for example [1]. Why do you not acknowledge this? It could be that you never applied for such data (family history) even if the data is available. Was this the case? Please clarify. Why not use it?

1. Jani BD, Nicholl BI, Hanlon P, Mair FS, Gill JM, Gray SR, Celis-Morales CA, Ho FK, Lyall DM, Anderson JJ, Hastie CE, Bailey ME, Foster H, Pell JP, Welsh P, Sattar N. Family history of diabetes and risk of SARS-COV-2 in UK Biobank: A prospective cohort study. *Endocrinol Diabetes Metab.* 2021 Oct;4(4):e00283. doi: 10.1002/edm2.283. Epub 2021 Jul 11. PMID: 34505416; PMCID: PMC8420405.

[ED: Please do clarify whether you can do this additional analysis.]

Reviewer 2:

The manuscript has improved following substantial revision and adding text on the role of genetic factors for observed associations, as well as lack of data on gestational age in the UK Biobank.

What should be further revised is that you mix up lack of data on genetics (that is available in UK Biobank, see many previous publications) with lack of data on family history (that you mention is not available) which covers a wider range of risk markers/traits/habits than only genetics (the so called "missing heritability"). However, I found evidence for that family history is in fact available in the UK Biobank questionnaire dataset, as in most other cohort studies, see link: <https://biobank.ndph.ox.ac.uk/ukb/label.cgi?id=100034>

This has also been documented in some publications, for example [1]. Why do you not acknowledge this? It could be that you never applied for such data (family history) even if the data is available. Was this the case? Please clarify. Why not use it?

1. Jani BD, Nicholl BI, Hanlon P, Mair FS, Gill JM, Gray SR, Celis-Morales CA, Ho FK, Lyall DM, Anderson JJ, Hastie CE, Bailey ME, Foster H, Pell JP, Welsh P, Sattar N. Family history of diabetes and risk of SARS-COV-2 in UK Biobank: A prospective cohort study. *Endocrinol Diabetes Metab.* 2021 Oct;4(4):e00283. doi: 10.1002/edm2.283. Epub 2021 Jul 11. PMID: 34505416; PMCID: PMC8420405.

[ED: Please do clarify whether you can do this additional analysis.]

Response: Thanks for these useful comments. We have considered family history (mother/father/sibling) of several major diseases (heart disease, stroke, cancer, chronic bronchitis/emphysema, high blood pressure, diabetes, Alzheimer's disease/dementia, Parkinson's disease, and severe depression) those were collected in the UK Biobank by: (1) including these variables in the fully adjusted model in the sensitivity analysis and (2) conducting subgroup analyses stratified by family history of these diseases. When additionally adjusting family history of several major diseases in the model, the

associations of birth weight and childhood body size with multimorbidity were similar to the main results (**Supplementary Table S10**). Moreover, such association was consistent among participants with or without family history of most diseases (**Supplementary Figure S4**). These results suggested that the association of birth weight and childhood body size with multimorbidity may be independent of familial genetic factors. Corresponding descriptions and discussions were added in the manuscript:

Methods:

“Third, we repeated the primary analyses by 1) excluding participants with missing data of covariates; 2) imputing missing data of covariates using multiple imputation for five times; and 3) including family history (mother/father/sibling) of several major diseases in the fully adjusted model: heart disease, stroke, cancer, chronic bronchitis/emphysema, high blood pressure, diabetes, Alzheimer’s disease/dementia, Parkinson’s disease, and severe depression (**Supplementary Table S2**). Fourth, subgroup analyses were performed stratified by age at baseline, sex, Townsend deprivation index, education level, and family history of major diseases. Potential effect modifications were evaluated by adding an interaction term of each variable and birth weight or childhood body size in the model.” (**Line 157-163, page 7**)

Results:

“When additionally adjusting family history of several major disease in the model, similar results were found (**Supplementary Table S10**). Moreover, the associations of birth weight and childhood body size with multimorbidity were consistent among participants with or without family history of most major diseases (**Supplementary Figure S4**).” (**Line 226-231, page 10**)

Discussion:

“When considering family history of several chronic diseases in the model, the associations of birth weight and childhood body size with multimorbidity were similar to the main results. Also, such association was consistent among participants with or

without family history of most diseases, implying that the association of birth weight and childhood body size with multimorbidity may be independent of familial genetic factors. According to a Mendelian randomization study which provided evidence of genetic pleiotropy between birth weight and cardiometabolic risk factors, maternal intrauterine environment, in which maternal genetic variation influences offspring birth weight, was not the major predictor of offspring cardiometabolic outcomes, but birth weight determined by the individuals' own genetic factors was more likely to associate with cardiometabolic outcomes¹. This study, coupled with our results, provided converging evidence that familial genetic effects that predispose to offspring birth weight, were not the main determinant of some chronic diseases and multimorbidity in an individual's later life. Future studies using genetic data are warranted to verify such points.” (Line 301-314, page 13-14)

Reference:

1. Moen GH, Brumpton B, Willer C *et al.* Mendelian randomization study of maternal influences on birthweight and future cardiometabolic risk in the HUNT cohort. *Nat Commun* 2020;**11**:5404.

REVIEWERS' COMMENTS:

Reviewer #2 (Remarks to the Author):

The manuscript has now improved substantially after adjustment also for family history as a proxy of genetic influences. Thank you!

Reviewer #2 (Remarks to the Author):

The manuscript has now improved substantially after adjustment also for family history as a proxy of genetic influences. Thank you!

Response: Thank you so much for reviewing our manuscript and for giving us positive and insightful comments!